# Better Set Representations For Relational Reasoning

**Qian Huang** *
Cornell University
qh53@cornell.edu

**Horace He** *
Cornell University & Facebook
hh498@cornell.edu

**Abhay Singh**
Cornell University
as2626@cornell.edu

**Yan Zhang**
University of Southampton
yz5n12@ecs.soton.ac.uk

**Ser-Nam Lim**
Facebook AI
sernam@gmail.com

**Austin R. Benson**
Cornell University
arb@cs.cornell.edu

## Abstract

Incorporating relational reasoning into neural networks has greatly expanded their capabilities and scope. One defining trait of relational reasoning is that it operates on a set of entities, as opposed to standard vector representations. Existing end-to-end approaches for relational reasoning typically extract entities from inputs by directly interpreting the latent feature representations as a set. We show that these approaches do not respect set permutational invariance and thus have fundamental representational limitations. To resolve this limitation, we propose a simple and general network module called Set Refiner Network (SRN). We first use synthetic image experiments to demonstrate how our approach effectively decomposes objects without explicit supervision. Then, we insert our module into existing relational reasoning models and show that respecting set invariance leads to substantial gains in prediction performance and robustness on several relational reasoning tasks. Code can be found at github.com/CUAI/BetterSetRepresentations.

## 1 Introduction

Modern deep learning models perform many tasks well, from speech recognition to object detection. However, despite their success, a criticism of deep learning is its limitation to low-level tasks as opposed to more sophisticated reasoning. This gap has drawn analogies to the difference in so-called "System 1" (i.e., low-level perception and intuitive knowledge) and "System 2" (i.e., reasoning, planning, and imagination) thinking from cognitive psychology [3, 9]. Proposals for moving towards System 2 reasoning in learning systems involve creating new abilities for composition, combinatorial generalization, and disentanglement [2, 13, 17].

One approach for augmenting neural networks with these capabilities is performing relational reasoning over structured representations, such as sets or graphs. This approach is effective in computer vision for tasks such as visual question answering, image captioning, and video understanding [7, 15, 19]. For relational reasoning, these systems are commonly split into two stages: (1) a perceptual stage that extracts structured sets of vector representations, intended to correspond to entities from the raw data, and (2) a reasoning stage that uses these representations. As the underlying data is unstructured (e.g., images or text), designing end-to-end models that generate set representations is challenging. Typical differentiable methods directly map the input to latent features using a feedforward neural network and partition the latent features into a set representation for downstream reasoning [1, 10, 15, 22].

However, this class of existing methods has a fundamental flaw that prevents them from extracting certain desired sets of entities — the so-called responsibility problem [24]. At a high level, if there

---

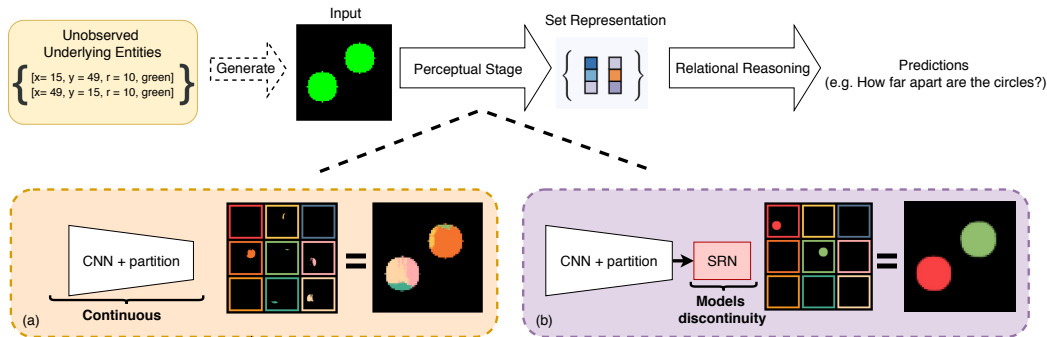

Figure 1: Overview of our Set Refiner Network (SRN) and the responsibility problem. The top row shows an overview of the relational reasoning paradigm. Under the assumptions of Theorem 1, a perceptual stage that effectively recovers the underlying set structure must be discontinuous. The second row shows a visualization of the perceptual stage for (a) existing methods and (b) the SRN on a reconstruction task. Each of the 9 color-coded panels corresponds to one of the 9 set elements. As existing methods (a) can only represent continuous functions, it is not able to recover the underlying object structure of the image. However, the SRN (b) allows us to model discontinuities and thus can recover the underlying object structure.

exists a continuous map that generates inputs from sets of entities, then any function that can map such inputs to a list representation of the entities must be discontinuous (under a few assumptions that we formalize in Section 2). For relational reasoning tasks, this implies the perceptual stage must contain discontinuous jumps. Existing methods use feedforward neural networks to approximate such a discontinuous map, so entities from certain inputs cannot be represented faithfully as shown in Figure 1. We demonstrate this problem extensively in Sections 2.2 and 3.1.

Here, we introduce the *Set Refiner Network (SRN)*, a novel module that iteratively refines a proposed set using a set encoder to manage the aforementioned responsibility problem. The main idea of our module is that *instead of directly mapping an input embedding to a set, we instead search for a set representation that can be encoded to the input embedding*. This search for a set is implemented via an iterative inference procedure, which is a better model for the discontinuous jumps required to address the responsibility problem. As shown in Figure 1, the SRN can produce set representations that are "properly decomposed," meaning that each set element corresponds to the entirety of one underlying entity or nothing at all. In extensive experiments, we demonstrate that the SRN can effectively decompose entities in a controlled setting (Section 2.2).

Furthermore, we hypothesize that proper decomposition obtained via SRNs can improve both the accuracy and robustness of downstream reasoning modules, in line with prior work which suggests that disentanglement is desirable for downstream tasks [17]. Intuitively, with a proper decomposition of entities, a reasoning module only needs to learn relations between two set elements to reason about two entities. On the other hand, if an entity is split amongst multiple set elements, the relational reasoning module needs to learn more complex relationships to perform the same task.

Indeed, we find that incorporating our SRN into relational reasoning pipelines in computer vision, reinforcement learning, and natural language processing tasks improves predictions substantially. On the Sort-of-CLEVR [15] benchmark, simply adding the SRN to a Relational Network reduces the relative error by over 50% on the most challenging question category. We also show that the SRN can improve the robustness of these relational reasoning modules. For example, even though a Relational Network has high test accuracy on easy questions within Sort-of-CLEVR, we show that the learned model is startlingly brittle, producing incorrect answers when the input is perturbed in a way that should not affect the answer. Again, simply plugging our SRN into the Relational Network resolves this issue, increasing robustness significantly.

## 1.1 Related Work

**Iterative Inference.** Iterative inference is the general idea of optimizing a learning procedure to recover latent variables instead of directly mapping from the input to the latent variables in one

step [12]. This idea has been used to predict sets in unsupervised object detection [6], general set prediction [23], and energy-based generative models [14]. Indeed, previous work has also noted that mapping from a set to a list requires iterative inference to avoid the responsibility problem [23, 24]. However, we are the first to demonstrate that (1) using iterative inference resolves a fundamental issue in relational reasoning over raw data, and (2) the ability for such an approach to learn objects *without direct supervision or reconstruction as an objective*. This second point prevents previous approaches from easily integrating into existing relational reasoning models. For instance, deep set prediction networks [23] require an additional loss term that uses the ground truth sets during training, which makes them unsuitable for our setting where ground truth sets are not available.

**Pre-trained Modules for Perceptual Reasoning.** An alternative method for perceptual reasoning is to use a pre-trained object detector [19, 20, 21]. An immediate disadvantage of such an approach is that it requires a comprehensive object detector, which might require extensive training data in a new application. However, even with a detector, this approach is limited by a priori imposing what "objects" should be, which may be unsuitable for different reasoning tasks. For example, should a crowd of humans be considered as one object, multiple, or the background? This might depend on the task. For these reasons, we only consider fully differentiable, end-to-end methods in this paper.

**Unsupervised Object Detection.** Unsupervised object detection also focuses on object representations from unstructured data [4, 6, 11]. These methods effectively decompose scenes without explicit supervision, but they often require sophisticated architectures specific to object detection and segmentation that are difficult to reuse [6, 18]. Similar to pre-trained object detectors, these approaches also manually impose a notion of what "objects" should be. In contrast, we propose an isolated module which can easily be inserted into many existing architectures for diverse applications.

## 2 Set Refinement Networks for Better Set Representations

The general model for relational reasoning with set-structured representations maps an unstructured data point $X$ (such as an image) to a set of vectors, which acts as an intermediate set representation of entities (e.g., objects). After, some model uses the set of vectors to make a prediction. We write this two-stage pipeline as follows:

$$S = G(X) \implies \hat{Y} = F(S). \tag{1}$$

Here, $G$ is a set generator function that maps an input $X$ to a set of vectors $S$, and $F$ is a (usually permutation-invariant) function that maps the set $S$ to a prediction $\hat{Y}$. We will typically think of $F$ as a differentiable relational reasoning module. As a concrete example, in our experiments with the Sort-of-CLEVR benchmark in Section 3.1, $X$ is an image with several entities (colored shapes), $G$ is based on a grid partition of a CNN's feature maps, and the function $F$ tries to answer questions of the form: what is the shape of the entity that is furthest away from the red circle?

With images and reasoning tasks, it is essential that the set $S$ is an effective representation for the reasoning module to make good predictions. Indeed, a proper disentanglement of the input into meaningful entities is crucial for performance in a variety of downstream tasks [17]. However, as previously mentioned, existing models suffer from a "responsibility problem," so it is difficult to decompose the input $X$ into a meaningful set of entities represented by $S$. We now formalize this responsibility problem.

**Theorem 1.** *Let $\mathcal{S}_n^d$ be the set of all sets consisting of $n$ $d$-dimensional vectors with $d, n \geq 2$. Assume that there exists a function $g \colon \mathcal{S}_n^d \to \mathcal{X} \subseteq \mathbb{R}^k$ that generates input data from sets, where $d$ is continuous with respect to the symmetric Chamfer and Euclidean distances. Consider any function $h \colon \mathcal{X} \to \mathbb{R}^{n \times d}$. If for every $V = \{v_1, \ldots, v_n\} \in \mathcal{S}_n^d$, there exists a permutation $\sigma$ such that $h(g(\{v_1, \ldots, v_n\})) = [v_{\sigma(1)}, v_{\sigma(2)}, ..., v_{\sigma(n)}]$, then $h$ is discontinuous.*

*Proof.* For sake of contradiction, suppose a continuous $h$ exists, then $h \circ g$ is an exact counter example for the responsibility problem theorem in Zhang et al. [24]. Thus, such a function cannot exist. ☐

In other words, if our data can truly be generated from underlying entities by a set function $g$, and $h$ perfectly captures the entities (in the sense that $h$ maps the input data to a list representation of the underlying entities), then $h$ must be discontinuous. For one example, partitioning an embedding of a feedforward network is a common way to construct a function $h$, which forces $h$ to be continuous.

---

**Algorithm 1** One forward pass of the Relational Reasoning System with SRN

---
1: $S_0 = G(X)$                             ▷ Encode input with the traditional perceptual stage $G$
2: $z = H_{\text{embed}}(X)$                         ▷ Begin SRN
3: **for** $i = 1...r$ **do**
4:      $L(S_i) = \|H_{\text{agg}}(S_{i-1}) - z\|_2^2$
5:      $S_i = S_{i-1} - \alpha \frac{\partial L}{\partial S_{i-1}}$                       ▷ Gradient step
6: **end for**
7: $S = S_r$                              ▷ End of SRN
8: $\hat{Y} = F(S)$                         ▷ Final prediction

---

In these settings, the distinction between $h$ in Theorem 1 and $G$ in Eq. 1 is merely that $G$ is the function which treats the "list representation" produced by $h$ as a set, i.e., the elements of $G(X)$ are the columns of $h(X)$. Such approaches cannot capture the necessary discontinuity in modeling the behavior of a set-based data generator. See the appendix for more discussion and extensions of this result.

In the rest of this section, we develop a general technique, which we call a *Set Refiner Network*, that can be "tacked on" to any set generator function $G$ in Eq. 1 to create better set representations through better modeling of the discontinuity. We then show in Section 3 how this improves performance in a variety of relational reasoning tasks that can be expressed via the pipeline in Eq. 1.

### 2.1 Methodology

Existing methods of implementing $G$ by partitioning latent features suffer from the responsibility problem, so the generated set representation is not always properly decomposed. Our main idea is to take the output $S_0$ of the set generator $G$ and "refine" $S_0$ to a better decomposed set $S$. We do so by iteratively improving $S_0$ so that it *encodes* to the original input.

Given a data point $X$, let $S_0 = G(X)$. We also generate a data embedding $z = H_{\text{embed}}(X)$ as a reference for what information should be contained in $S$. In practice, we implement $H_{\text{embed}}$ by sharing weights with $G$, e.g., by taking a linear combination of intermediate feature maps (Section 3.1).

We then address the responsibility problem with an iterative inference approach, in line with recent research [6, 18, 23]. In particular, given the data point embedding $z$, we seek to find a set $S$ *that encodes to $z$*, i.e., we want to solve the optimization problem

$$S = \arg\min_{S'} L(S') = \arg\min_{S'} \|H_{\text{agg}}(S') - z\|_2^2, \tag{2}$$

where $H_{\text{agg}}$ is a permutation-invariant set encoder function parameterized by neural networks. Finding a set this way forces permutation invariance into the representation directly, as $H_{\text{agg}}$ is a permutation-invariant set function, while also ensuring that the set captures information in the embedding $z$.

In order to train in an end-to-end fashion, we use what we call a *Set Refiner Network (SRN)*, which is defined by the following inner optimization loop:

$$S = \text{GradientDescent}_{S'}(L(S'), S_0, r), \tag{3}$$

which means that we run gradient descent for the loss in Eq. 2 for $r$ steps, starting from the initial point $S_0$. Collectively, this produces the set input $S$ for the function $F$ making predictions for the downstream task (Alg. 1). We can also view this entire procedure as a new perceptual stage $G'$.

An alternative view of the SRN is as a generative model akin to representation inversion [5] or energy-based models [14]. Given a label and an image classifier, for example, one can generate an image that matches a label through iterative inference. In our case, given an embedding $z$ along with a set encoder $H_{\text{agg}}$ that maps a set to an embedding, we generate a set that matches the embedding through iterative inference.

The choice of the module $F$ in Eq. 1 is crucial for the SRN to be able to decompose the latent space into meaningful entities. In particular, $F$ should be a set function that ignores the order of objects, which is the case in tasks such as relational reasoning. As in previous work [10, 18], we expect that this symmetry assumption will force the network to learn a decomposition. Intuitively, pushing all information into one set element disadvantages the model, while all set elements need to contain

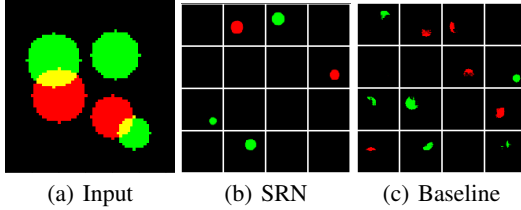

| (a) Input | (b) SRN | (c) Baseline |

Figure 2: Decomposition results of SRN (b) and Baseline (c) models over a sample input (a) in the image reconstruction task with the synthetic Circles Dataset. The SRN successfully decomposes the image. Appendix B has additional examples.

Table 1: Decomposition success rate on the Circles Dataset. Including the SRN drastically improves the decomposition of the images into its constituent entities.

|  | % of Images Decomposed | |
|---|---|---|
| # of Circles | Baseline | SRN |
| 1 | 45.1% | **95.8**% |
| 2 | 16.5% | **88.7**% |
| 3 | 7.8% | **79.9**% |
| 4 | 3.2% | **72.7**% |
| 5 | 1.9% | **65.2**% |

similar "kinds" of information as they are processed identically. Despite this assumption, typical set generation processes still fail to enforce permutation invariance, which is fixed by including our SRN.

Note that our choice of $F$ is similar to DSPN [23]. The primary difference is that SRN removes DSPN's dependency on ground truth sets during training. This is a fundamental change — DSPN cannot be used in settings other than supervised set prediction, as noted in the paper: "when naively using [the image embedding] as input... our decoding process is unable to predict sets correctly from it. To fix this, we add a term to the loss [that depends on ground truth sets]". SRN addresses this limitation by refining a predicted set instead of using a shared initialization across all inputs, which also allows easy integration into existing relational reasoning pipelines.

**Graph Refiner Network (GRN) extension.** We can extend the SRN technique to other types of structured representations. Here, we extend it to graph representations. In this case, the function $G$ in Eq. 1 produces a graph instead of a set, and the downstream reasoning module $F$ operates on a graph. The only change we need to make is to replace the set encoder $H_{\text{agg}}$ in SRN with a graph encoder. To do this, we use a Graph Neural Network [2] that creates a vector representation of a graph. Furthermore, even if $G$ produces a set, we can still interpret this to be a graph where the nodes correspond to the set elements and the edges are initialized to some pre-configured initialization. We consider the complete graph as an initialization, where the refinement step seeks to find weights on the edges in the complete graph that are useful for a downstream reasoning task. We call this approach a *Graph Refiner Network (GRN)*. However, we find that the GRN is difficult to train and usually does not lead to much improvement over SRN, and we only use the GRN in Section 3.3.

## 2.2 Explanatory experiments

Before turning to more sophisticated learning pipelines in Section 3, we first consider the simpler task of image reconstruction to demonstrate the capabilities of our Set Refiner Networks. Image reconstruction is a useful sandbox for understanding set representations which will aid intuition for more complex reasoning tasks. The set representations should contain as much information about the original image as possible and the latent structure can be easily visualized as a set of objects.

To this end, we construct a synthetic "Circles Dataset" for easy control over the latent structure. Each image is $64 \times 64$ pixels with RGB channels in the range 0 to 1 (Fig. 2(a) is an example data point). An image contains 0 to 10 circles with varying color and size. Each circle is fully contained in the image with no overlap between circles of the same color. We use 64000 images for training and 4000 images for testing.

We compare two models that implement the basic pipeline in Eq. 1. The models use the same set function $F$ but different set generators $G$. First, the *Baseline model* implements $G$ using a CNN with groups of channels of the final feature map interpreted as set elements; this follows the approach of differentiable object extractors [1, 10]. Second, the *SRN model* extends the baseline $G$ by adding the SRN as in Alg. 1. Specifically, $H_{\text{embed}}$ is implemented by flattening the final feature map and passing it through a linear layer, and $H_{\text{agg}}$ is a set encoder that processes each element individually with a 3-layer MLP, followed by FSPool [24]. The set function $F$ decodes each element to an image $I_i$ independently through shared transposed-convolution layers, finally summing the generated images,

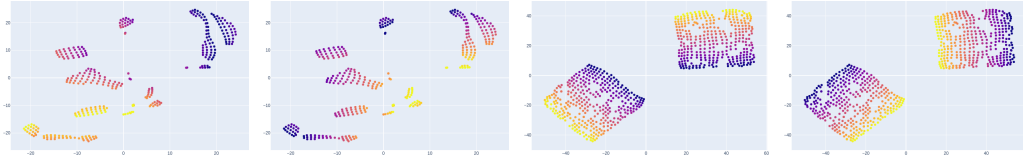

| (a) Baseline X coordinate | (b) Baseline Y coordinate | (c) SRN X coordinate | (d) SRN Y coordinate |

Figure 3: t-SNE plots of latent set elements. Points are colored by X and Y coordinate values in (a), (c) and (b), (d). The SRN representations smoothly vary with coordinates and has two distinct clusters corresponding to two colors.

weighted by their softmax score to ensure their sum lies in $[0, 1]$:

$$I_i = \text{TransposeConvs}(S_i), \quad \hat{Y} = F(S) := \sum_{i=1}^{|S|} \text{softmax}_i(I) \cdot \text{sigmoid}(I_i). \quad (4)$$

We train the model with squared error image reconstruction loss using the Adam optimizer with learning rate 3e-4. See Appendix B for the full architecture details.

Figure 2 shows an example reconstruction and the images decoded from the set elements. Although we only provide supervision on the entire image's reconstruction, SRN naturally decomposes most images into a set of the individual circles. In contrast, the baseline approach does not decompose the circles properly (Fig. 2(c)). We also quantitatively measure the quality of the set representations in terms of how well they decompose the circles into separate set elements. We say that a decomposition of an image is *successful* if the number of set elements containing objects is equal to the number of circles in the image, where a set element is considered to contain an object if the decoded image has an $L_{\text{inf}}$ norm greater than 0.1. Using this metric, we find that the SRN is far more successful at decomposition than the baseline (Table 1), especially when there are multiple circles. We repeat the same experiments with the CLEVR dataset [8] and observe similar results (see Appendix C).

Interpolating circles from one position to another illustrates why we see this difference in decomposition. Both models enforce symmetric processing of set elements, which causes both models to exhibit some form of decomposition. However, as the baseline is unable to overcome the responsibility problem, it cannot use the natural decomposition of the image into objects and instead must decompose the image by location. As we move the circles from left to right, the baseline gradually hands responsibility for the circle from one set element to the other. This causes failures of proper decomposition *in between* the two locations. On the other hand, the SRN can discontinuously shift responsibility from one set element to another, thus properly decomposing the image throughout the interpolation. Appendix B has a visualization of this behavior.

Finally, we visualize how the SRN produces meaningful structure in the latent space. We grid-sample images with one red or green circle at different locations and plot the latent space of the set elements corresponding to the circle using a two-dimensional t-SNE embedding with default scikit-learn settings (Fig. 3). The set elements generated by the baseline are in discontinuous clusters, while the SRN shows the desired grid structure with two clear factors of variation — the X and Y coordinates of the circles, with two clusters for the two colors. We also visualize coordinates, radius, and color with a three-dimensional t-SNE embedding in Appendix B, with similar results. Again, this discrepancy is due to the responsibility problem, and the smooth latent space of SRN implies that the model has properly decomposed objects from multi-object images.

## 3 Relational Reasoning Experiments

In this section, we plug our SRN module into three diverse reasoning tasks: reasoning about images, reasoning within reinforcement learning, and reasoning about natural language. In all cases, we see that including the SRN refinement step within an existing reasoning pipeline leads to a substantial increase in performance. Moreover, we show how the SRN can make reasoning modules more robust.

### 3.1 Relational Reasoning with Images from Sort-of-CLEVR

So far, we have shown that the SRN can perform unsupervised object detection in a reconstruction setting, while the baseline cannot. We now demonstrate that this advantage can improve both

Table 2: Sort-of-CLEVR performance (mean accuracy and standard deviation over five runs).

| | Performance | | |
| Task | RN | RN + SRN | % Error reduction |
| --- | --- | --- | --- |
| Relational | $95.1_{\pm 0.4}$ | $\mathbf{96.9}_{\pm 0.3}$ | 37 |
| Furthest | $91.4_{\pm 0.5}$ | $\mathbf{95.9}_{\pm 0.4}$ | 52 |
| Closest | $93.9_{\pm 0.9}$ | $\mathbf{94.8}_{\pm 0.5}$ | 15 |
| Count | $\mathbf{100}_{\pm 0.0}$ | $\mathbf{100}_{\pm 0.0}$ | N/A |
| Robustness (Synthetic) | $1.5_{\pm 1.2}$ | $\mathbf{96.7}_{\pm 2.3}$ | 97 |
| Robustness (Dataset) | $61.3_{\pm 6.2}$ | $\mathbf{93.2}_{\pm 3.6}$ | 82 |

performance and robustness on a relational reasoning task. To this end, we use the Sort-of-CLEVR task [15], which is designed to test relational reasoning capabilities of neural networks. The dataset consists of images of 2D colored shapes along with questions and answers about the objects. The questions are split into non-relational questions and relational questions, and the relational questions are further split into questions of the form "What is the shape of the object that is the furthest from the red object?"; "What is the shape of the object that is the closest to the grey object?"; and "How many objects are the same shape as the green object?" We call these 3 question categories *Furthest*, *Closest*, and *Count*. We use 100000 images for training, 1000 for validation, and 1000 for test. Appendix D has additional experimental details.

**Model.** We use the same RN architecture as Santoro et al. [15] and insert our SRN between the perceptual end (i.e, the convolutional feature extractor) and the Relational Network, similar to the reconstruction experiment in Section 2.2. All other network and training parameters are kept constant. We train all models for 50 epochs and select the epoch with the best validation accuracy.

**Results.** For all experiments, we run 5 trials and report the mean and standard deviation (Table 2). On the most challenging question category (Furthest task), we improve the performance from 91.4% to 95.9%, reducing the error by 52%. We note that this is the category that requires the most global reasoning from the network. We also demonstrate slight improvements on the Closest task. Overall, we increase average performance from 95% to 97%.

In order to verify the importance of the inner optimization loop, we take an SRN model trained with 5 inner optimization steps and evaluate with 0 steps. The Count task performance stays at 100%, the Closest task performance drops modestly from 95% to 87%, and the Furthest task drops substantially from 96% to 63% (random guessing achieves 50%). This is evidence that the inner optimization step is necessary for the furthest task, which is the one with the most global reasoning.

**Robustness.** To verify our original hypothesis about the fundamental representation discontinuity with Relational Networks, we define two robustness tasks. In the first one, which we call the "Synthetic" task, we construct an "easy" input configuration on which both models provide the correct answer with very high confidence, and create our dataset by translating it to 200 different positions (Appendix D has more details on the setup). We test on the "Furthest" task, which we believe requires the most global relational reasoning. For a given image, we define the model to be robust on that image if the model provides the correct answer on all of 720 evenly spaced rotations. We rotate the shape coordinates and not the image itself, as all shapes during training are axis-aligned. We find that the baseline RN is robust on only 1.5% on this synthetic robustness dataset, while the model with SRN achieves 96.7% robustness (Table 2).

To demonstrate generalization in a setting closer to the training data, in our second robustness task ("Dataset" task), we generate 1000 new "easy" images as a test set, where an image is "easy" if the furthest shape is at least twice as far as the second furthest shape. Both models achieve >99% accuracy on this subset. However, the RN is robust on only 61% of images, while the RN augmented with SRN is robust on 93% of images. Finally, in order to verify that robustness is not simply a result of better overall performance, we evaluate the SRN at an early stage of training on the robustness tasks (Epoch 6). At this point, the SRN's relational accuracy is lower than the RN accuracy at the end of training (93.7% vs. 95.1%). Still, the SRN performs better on both the Synthetic and Dataset robustness tasks (55% and 84% respectively).

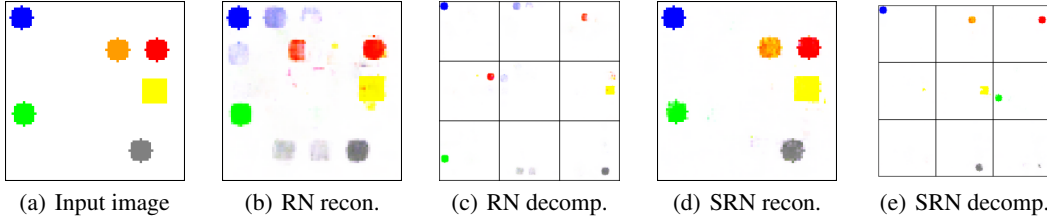

| (a) Input image | (b) RN recon. | (c) RN decomp. | (d) SRN recon. | (e) SRN decomp. |

Figure 4: Sort-of-CLEVR reconstruction and decompositions (top 9/25 elements ranked by $\ell_1$ norm). Not only is the baseline RN unable to reconstruct the image properly, it is also unable to decompose the objects, while the SRN properly decomposes the image and reconstructs the image.

**Visualization.** Unlike the image reconstruction task, there is no easy way to visualize the object representations. Thus, for both RN and RN+SRN, we freeze the perceptual stage (i.e., set generator) and replace the relational module with the image reconstruction module in Section 2.2. We overfit on a dataset of 10000 images; this can be viewed as capturing what meaningful information the latent set elements can contain. The number of trainable parameters is identical for both models.

The quality of the SRN reconstruction is far better than the RN reconstruction. After 50 epochs, the baseline RN has 5000 squared error reconstruction loss on the training set while SRN has 773. One reason for this is that the RN will often "hallucinate" objects that do not exist (Figure 4). For example, even though all input images have one blue shape, the RN will occasionally generate two or three blue shapes. This suggests that the responsibility problem shows up, as the baseline RN splits objects into multiple set elements. As each set element is decoded independently and identically, the image decoder is unable to decide which element is the actual circle, producing the hallucinations.

## 3.2 World Model Learning

We demonstrate the advantage of our approach in a more complicated setting, *world model learning*, where the goal is to learn and predict state transitions in a Reinforcement Learning environment.

**Model.** We plug our SRN module into C-SWM [10], which uses relational reasoning as the core system to learn a world model. In C-SWM, the object representations are extracted by interpreting each channel of the feature maps as an object mask and then passing them through the object encoder individually. The resulting "state vectors" are intended to encode the abstract state of each object, which are then used in the Relational Transition Model. This perceptual stage design is similar to the baseline in our image reconstruction experiment in Section 2.2. We plug in our SRN to refine the set of state vectors, with the original state vectors as the initial guess $S_0$ and a linear projection of object masks as the embedding $z$. The number of set elements is set to be the same as the C-SWM hyperparameter $K$. We sweep learning rates for both models and trained for 500 epochs, which led to better baseline performance compared to the original implementation of C-SWM. More training and architecture details are in Appendix E.

**Results.** We see performance improvements by incorporating the SRN into C-SWM on the Atari Pong (Pong) and Space Invaders (Space) datasets (Table 3). These are the only 2 tasks for which C-SWM does not achieve near 100% performance. As the original paper [10] notes, one limitation of C-SWM is that the model cannot disambiguate multiple identical objects and that some kind of iterative inference procedure would be required to resolve this limitation. Thus, it is perhaps unsurprising that including the SRN provides especially better improvements for the Space dataset, which contains several identical objects. In addition, using the SRN has especially notable improvements after 10 action steps.

## 3.3 Relational Reasoning from Natural Language

Finally, we show that our approach is also applicable over inputs other than images through the text-based relational reasoning task CLUTRR [16]. In this task, the reasoning task is inferring kinship relationships between characters in short stories that describe the family tree. We use the $k = 2, 3, 4$ datasets [16] as the training set. We plug in both SRN and the graph version (GRN) into the RN baseline with LSTM encoders. To adapt to the various number of sentences across the dataset, we

Table 3: C-SWM hits at rank 1 [10] and mean reciprocal rank accuracy with and without SRN on the Atari Pong (Pong) and Space Invaders (Space) datasets (mean and standard deviation over 10 trials). The number in the parenthesis after model names is the number of object slots $K$.

| | Model | 1 Step | | 5 Steps | | 10 Steps | |
|---|---|---|---|---|---|---|---|
| | | H@1 | MRR | H@1 | MRR | H@1 | MRR |
| Pong | C-SWM(5) | **63.3** $_{\pm 1.6}$ | **75.6** $_{\pm 1.0}$ | 29.8 $_{\pm 1.5}$ | 46.1 $_{\pm 1.4}$ | 16.9 $_{\pm 2.0}$ | 31.5 $_{\pm 2.2}$ |
| | SRN (5) | 62.9 $_{\pm 1.4}$ | 75.4 $_{\pm 0.8}$ | 30.3 $_{\pm 0.8}$ | 46.5 $_{\pm 0.7}$ | **19.8** $_{\pm 1.2}$ | **35.3** $_{\pm 0.9}$ |
| | C-SWM (3) | 57.3 $_{\pm 5.0}$ | 71.6 $_{\pm 3.3}$ | 28.0 $_{\pm 1.3}$ | 44.3 $_{\pm 1.5}$ | 14.8 $_{\pm 1.8}$ | 29.4 $_{\pm 1.9}$ |
| | SRN (3) | 62.7 $_{\pm 2.5}$ | 75.3 $_{\pm 1.6}$ | **31.7** $_{\pm 2.2}$ | **48.2** $_{\pm 2.0}$ | 17.1 $_{\pm 2.5}$ | 32.1 $_{\pm 2.9}$ |
| Space | C-SWM(5) | 53.8 $_{\pm 3.4}$ | 70.7 $_{\pm 2.7}$ | 50.2 $_{\pm 1.6}$ | 68.1 $_{\pm 1.2}$ | 28.7 $_{\pm 2.5}$ | 46.6 $_{\pm 3.1}$ |
| | SRN (5) | 63.6 $_{\pm 3.2}$ | 78.2 $_{\pm 2.3}$ | 59.9 $_{\pm 5.3}$ | 75.3 $_{\pm 3.9}$ | 39.5 $_{\pm 3.4}$ | 59.3 $_{\pm 2.8}$ |
| | C-SWM(3) | **69.0** $_{\pm 0.9}$ | **81.6** $_{\pm 0.6}$ | 60.9 $_{\pm 1.1}$ | 75.9 $_{\pm 0.8}$ | 38.1 $_{\pm 3.7}$ | 57.5 $_{\pm 3.9}$ |
| | SRN (3) | 68.4 $_{\pm 0.8}$ | 81.3 $_{\pm 0.4}$ | **65.0** $_{\pm 2.8}$ | **78.8** $_{\pm 2.0}$ | **47.8** $_{\pm 7.9}$ | **66.7** $_{\pm 6.6}$ |

fix the set size to 20, with LSTM sentence embeddings padded with zeros as the initial guess $S_0$. The average cell states over all sentences are used as the data embedding $z$. We use the original model, except that we add 0.5 encoder dropout and 1e-4 $\ell_2$-regularization (5e-6 for GRN) to avoid overfitting. Over 5 runs, for $k = 2, 3, 4$, the best average performance across epochs for the baseline is 49.7%, while including the SRN reaches 55.0%, and using the GRN reaches 56.1%.

## 4 Conclusion

We have provided substantial evidence that standard approaches for constructing sets for relational reasoning with neural networks are fundamentally flawed due to the responsibility problem. Our Set Refinement Network module provides a simple solution to this problem with a basic underlying principle: instead of mapping an input to a set, find a set that maps to (an embedding of) the input. This approach is remarkably effective and can easily be incorporated into a variety of learning pipelines. We focused on relational reasoning in this paper, as set structure is fundamental there, but we anticipate that our approach can be used in a variety of domains using set structure.

## Acknowledgments

We thank the rest of Cornell University Artificial Intelligence for helpful discussions, as well as CUAI advisors Professor Serge Belongie and Professor Kavita Bala for making this paper possible. In addition, we would like to thank Yewen Pu (evanthebouncy on Twitch) for suggestions on Figure 1. The academic co-authors of this research were supported by Facebook AI, NSF Award DMS-1830274, ARO Award W911NF19-1-0057, ARO MURI, and JP Morgan Chase & Co.

## Broader Impact

As neural networks continue to be deployed in a number of high-stakes tasks, it is crucial to have a better understanding of their limitations and robustness. In this paper, we make some progress on both of these issues. We showed that end-to-end learning systems that use an intermediate set-structured representation often have difficulty properly decomposing the input into set elements, illustrating that existing pipelines are not actually creating the representations that they intended to make. Also, our experiments in Section 3.1 demonstrated that a popular relational reasoning approach in fact learns a brittle model on a fairly simple dataset. We have presented a first approach at alleviating some of these systemic problems, and our experiments highlight how to learn more meaningful set representations, which helps improve robustness in addition to predictive performance.

In addition, our method improves reasoning capabilities of a wide variety of neural networks, which might increase the likelihood of such systems being used in the wild. Most modern machine learning systems are only relied upon to do "System 1" tasks, and we are contributing to models being relied upon for "System 2" tasks as well. This has many potential ramifications, both positive and negative, largely related to the general problems of using machine learning in practice.

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
