[Supplementary Material]

# Better Set Representations For Relational Reasoning
## Supplementary Material

**Qian Huang** *
Cornell University
qh53@cornell.edu

**Horace He** *
Cornell University & Facebook
hh498@cornell.edu

**Abhay Singh**
Cornell University
as2626@cornell.edu

**Yan Zhang**
University of Southampton
yz5n12@ecs.soton.ac.uk

**Ser-Nam Lim**
Facebook AI
sernam@gmail.com

**Austin R. Benson**
Cornell University
arb@cs.cornell.edu

## A    Theorem of the responsibility problem in relational reasoning

**Theorem 1.** *Assume that there exists a set function $g : \mathcal{E} \to \mathcal{X} \subseteq \mathbb{R}^k$ that generates input from set of entities ($\mathcal{E} = \mathcal{S}_n^d$ is the set of all sets of size n with elements in $\mathbb{R}^d$, $d \geq 2, n \geq 2$), and g is continuous with respect to the symmetric Chamfer and Euclidean distances. Let $h \colon \mathcal{X} \to \mathbb{R}^{n \times d}$. If for every $V = \{v_1, \ldots, v_n\} \in \mathcal{S}_n^d$, there exists a permutation $\sigma$ such that $h(g(\{v_1, \ldots, v_n\})) = [v_{\sigma(1)}, v_{\sigma(2)}, ..., v_{\sigma(n)}]$, then h is discontinuous.*

*Proof.* For sake of contradiction, suppose a continuous $h$ exists, then $h \circ g$ is an exact counter example for the responsibility problem theorem in [9]. Thus, such a function cannot exist.   □

Generalizations that we can make to help in application in practice:

1. $\mathcal{E}$ can be restricted to subsets of the whole space. One just needs to show the existence of any smooth curve $\gamma \subseteq \mathcal{E}$ that connects two sets with different permutation. This curve can then be used to replace the circle in the original proof. In the Circles Dataset, we can let $\gamma$ be the rotation of two green circles (while red circles are unchanged), such that $f$ must contain discontinuous jump. Such curve should commonly exist since entities, like input images, generally share the same support (or even assumed to be independent and identically distributed).

2. This theorem can also be applied by letting $\mathcal{E}$ and $\mathcal{X}$ be a subset of real data. For example, we can apply this theorem to the images with only green circles in the Circles Dataset, such that $f$ must contain discontinuous jump trivially. This reduces the assumptions on the existence of $g$ to only a subset of the dataset.

## B    Circles Dataset Reconstruction Experiment

### B.1    Architecture Details

**Set Generator** For the baseline, the set generator $G$ is a standard image encoder derived from Stacked Convolutional Auto-Encoders [5] and DCGAN [7] with additional processing at the end similar to [4] and [1]:

1. Conv2d layer takes 3 channels as input, 64 filters, 4 kernel size, stride 2, padding 1, no bias, relu activation.

---

2. Conv2d layer takes 64 channels as input, 128 filters, 4 kernel size, stride 2, padding 1, no bias,batch normalized, relu activation.

3. Conv2d layer takes 128 channels as input, 256 filters, 4 kernel size, stride 2, padding 1, no bias, batch normalized,relu activation.

4. Conv2d layer takes 256 channels as input, 512 filters, 4 kernel size, stride 4, padding 1, no bias,batch normalized, relu activation.

5. Reshape to ($|S|$, 2048 / $|S|$), where $|S|$ is the size of the set, and transpose dimension 1,2

6. Conv1d layer takes (2048 / $|S|$) channels as input, element dimension number of filters, 1 kernel size.

The output tensor would then be of shape $(b, n, |S|)$, where $b$ is the batch size and $n$ is the dimension of each set element. This output is interpreted as a set of vector elements for each instance, with the corresponding set size and element dimension pre-specified. This generator is designed to process images with the convolutional layers. The output of step 4 can be seen as a set of feature maps, where each feature map has equal perception field that covers the whole image. We then use step 5 and step 6 to transform this set to the desired shape by grouping feature maps and processing each feature map group individually with shared function. Overall, this makes sure that each set element is produced with the same initial input (the whole image) and with the same architecture, although the weights might be different.

For SRN, the same architecture is used for predicting the innital guess. After that, the embedding is generated by flattening the feature maps and projecting down to a 100-dimensional vector using a fully connected layer. $H_{\text{agg}}$ processes each element in the input set individually with a 3-layer MLP with 512 as the hidden dimensions, followed by FSPool [9] with 20 pieces and no relaxation.

**Set Function** As described in the main paper, the prediction function $F$ decodes each element in the generated set to an image independently through shared transpose-convolution layers. We then aggregate the set of images with self attention. Both baseline and SRN have the same image decoder (TransposeConvs) architecture:

1. Fully connected layer, projecting from element dimension $n$ to feature maps of shape (1024, 4, 4). Apply batch norm before reshape and use relu activation.

2. TransposeConv layer filters 512, kernel size 4, stride 2, padding 1, no bias, batch normalized, relu activation.

3. TransposeConv layer filters 256, kernel size 4, stride 2, padding 1, no bias, batch normalized, relu activation.

4. TransposeConv layer filters 3, kernel size 4, stride 4, padding 0, no bias.

This architecture first creates objects and then superimposes them for the final reconstruction. This is permutation invariant as the order of the elements in the set does not change the output and allows each element to be interpreted from the individual reconstructed objects, leading to direct disentanglement.

## B.2 Disentanglement Metric Specification

To measure whether an image was completely disentangled, we examine each of the individual images generated by each of the set elements. If any of the values were more than .1 (out of 1), we considered that set element to be "non-empty." If the total number of non-empty set elements matched the total number of circles, we considered the image to be "completely disentangled." Although it is possible that this results in false positives (for example, if the model splits one circle into 2 set elements and puts 2 other circles in one set element), we rarely observe this in practice.

## B.3 Interpolation plots

As described in the main paper, the baseline gradually hands responsibility for the circle from one set element to the other, as we move the circles from left to right. This is visualized in 1, where the colors represent the responsibilities as in the Figure 1 in the main paper.

Figure 1: Reconstruction results while input is gradually shifting to the right.

## B.4    3d t-SNE plots

The three-dimensional t-SNE plots with variations of coordinates, radius and color are shown in Figure 2.

(e) X coordinate        (f) Y coordinate        (g) Radius        (h) Color

Figure 2: Three-dimensional t-SNE embeddings of set elements generated by baseline (top) and SRN (bottom) method, colored by various parameters of the circles: X coordinate (a), Y coordinate (b), radius (c), and color (d). The representations have clear continuous structure.

## B.5    Additional Results

Figures 3 shows 10 randomly sampled images from the test set along with the latent sets learned by SRN. As shown in the figure, the decomposition is almost perfect for SRN, whereas baseline frequently cannot disentangle objects.

## C    CLEVR Object Reconstruction Results Experiment

**Dataset**    We use the CLEVR dataset [3] to show that such object decomposition holds in more complicated settings. The dataset contains 70,000 training and 15,000 validation images. The original dataset does not contain images of the masked objects, but we generate these through publicly available code.

**Model**    We again encode images using a CNN that takes $128 \times 128$ images as input and has four convolutional layers to produce a 512-dimensional image embedding. The architecture for the CLEVR experiment are nearly the same as ones for the Circles Dataset described above. The only

Table 1: Intersection over Union (IoU) results for CLEVR image reconstruction. Our SRN approach out-performs the MLP baseline.

| Model Description | Baseline | SRN |
|---|---|---|
| overall IoU | 0.9193 | **0.9345** |
| per-object IoU | 0.7737 | **0.8305** |

difference is that the projection layers used in image embedding generation and the decoder are modified to adapt to image size of $128 \times 128$.

**Results**   Figures 4 and 5 show 10 randomly sampled images from the test set, along with the SRN and baseline latent sets. Figures 6, 7, and 8 each show one sample in more detail. Again, SRN disentangled objects and completed occluded part of the objects reasonably, while the baseline failed to disentangle objects.

We also measured the decomposition quality using intersection over union (IoU) score (Table 1). We use two metrics for evaluation: the standard overall intersection-over-union (IoU) and a per-object intersection-over-union. The per-object IoU is the IoU over the Chamfer matching between ground-truth bounding boxes and the bounding boxes of the predicted decomposition. In both cases, we first threshold the pixels valued in $[0, 1]$ by $0.01$ (i.e., pixels with all channels smaller than $0.01$ are set to zero). For the overall IoU case, this threshold is applied on the final reconstruction, and in the per-object case it is applied on each set element. Due to rendering noise, it is difficult to obtain instance segmentation masks. Instead, we algorithmically generate bounding boxes with OpenCV's `findContours()` function applied to the thresholded prediction. These bounding boxes are then compared to ground truth bounding boxes. We match the latent set of a prediction (where each element may contain multiple bounding boxes if a prediction fails to disentangle objects) with the set of generated bounding boxes using Chamfer matching, where the cost is the IoU between the pairs of elements: each prediction is matched with the closest ground truth bounding box. With this computed assignment, we then compute the average per-object IoU.

The overall IoU is calculated pixel-wise over all objects in the foreground in the reconstructed image. The per-object IoU is the IoU over the Chamfer matching between ground-truth bounding boxes and the bounding boxes of the predicted decomposition. In both cases, SRN performs better than the baseline. The low per-object IoU for the baseline is due to the poor object decomposition and its inability to handle occluded objects. In contrast, SRN can decompose the scene with the superposition of full objects, including the occluded parts, providing a more meaningful disentangled representation.

# D   Sort-of-CLEVR

## D.1   Architecture Details

We reuse the Relational Network and perceptual end implementations from the most popular open source Relational Network implementation [2]. The architecture used is 4 convolution layers with 24 channels each, 3x3 kernels, 2 stride, and 1 padding. Between each convolution layer is a relu and a batchnorm layer.

This results in a (batch x filters x height x width) tensor (batch x 24 x 5 x 5) in our case. Each of the (5x5) cells in the feature map is treated as a size 24 feature map. These represent the "entities" which we perform relational reasoning on.

The question is encoded as a binary vector with 11 elements. The first 6 are an one hot encoding of which of the 6 colors the question is about. The next 2 are an one hot encoding of whether the question is relational or non-relational. The last 3 are an one-hot encoding of the 3 question subtypes.

The question is concatenated to each of the entities, and is passed to the Relational Network. The $g_\theta$ (as in [8]) is a 4 layer MLP with 256 elements per layer and ReLU non-linearities. A final Linear/ReLU layer is used at the end.

### D.2 Dataset Details

We use the dataset generator from the same source as the model. However, instead of 10k images we use 100k. With a lower dataset size, both models tend to overfit and make it more difficult to demonstrate the responsibility problem vs general inaccuracy. Our limited experiments still demonstrate that SRN still improves both robustness and performance on the smaller dataset as well.

The configuration used in Robustness (Synthetic) is shown in Figure 9. The yellow shape is randomly sampled between a circle/rectangle to avoid degenerate solutions like predicting the opposite of the red shape. All questions are of the form "What is the color of the furthest shape from the red shape?".

The dataset used in Robustness (Dataset) is generated with the same code as the one used for training, with 2 exceptions. First, we only keep "easy" images - in other words, images where the shape furthest away from the red shape is at least twice as far as the second furthest shape. Second, we enforce all shapes to not be within 15 of the border. Otherwise, the rotations required for checking "robustness" would move the shapes outside of the borders.

### D.3 Hyperparameters Considered

We find that this task tends to be fairly robust to hyperparameters. We tried both 5 steps and 10 steps for inner optimization steps, and did not find a significant difference. The inner learning rate is set to $0.1$ - changing the inner learning rate did not show a significant difference in performance either.

### D.4 Experiment with Cluttered Background

As an example where traditional approaches might have more difficulty, we try a task with a cluttered background. We fill the background with 5 background rectangles with random color, height, and width. Note that without a task, it would seem like there are now 11 "objects" in this scene. For the baseline RN, this task poses significantly more difficulty, even on the non-relational task. After 20 epochs, it reaches 98.5% accuracy on non-relational questions, and 94.0% accuracy on relational questions. On the other hand, the SRN still reaches 99.9% performance on non-relational questions, and 96.0% accuracy on relational questions.

## E World Model Learning

### E.1 Architecture Details

As described in the main paper, we used SRN module to refine the original state vectors $S_0$. Data embedding $z$ is generated by flattening the feature maps generated by object extractor, and passing it through a linear layer to 512 dimension (same as the number of hidden units in transition MLP). $H_{\text{agg}}$ is the same as in the circles reconstruction experiment.

### E.2 Training and Hyper-parameters Details

We use the official C-SWM code [3] and follow most of the training procedures in the paper, except for the evaluation set size, learning rate and number of epochs. We increase the evaluation set size from 100 to 1000 episodes to reduce variance. For learning rate, we test 5e-5, 1e-4, 5e-4, 1e-3, among which 1e-4 has substantially better performance. So we use 1e-4 learning rate for all runs and train until 500 epochs to reach convergence. The comparison of the baseline between original setup and new setup is included in table 2. Note that there is a large variance of performance across epochs. In the main paper table, In general, the epochs selected are all close to 500. For both datasets, all of our improvements on 5 and 10 steps are statistically significant ($p = 0.05$) with Welch's t-test. For example, on Space Invaders (3 objects, H@1, 10 steps), where SRN performance is most variable, a Welch's t-test provides a 95% confidence interval of 3.7% to 15.7% absolute improvement.

For Pong, SRN has inner learning rate $0.045$ for 3 objects and $0.0025$ for 5 objects, with $r = 10$. For Space Invader, SRN has inner learning rate $0.05$ and $r = 10$. We selected these values by starting with a random sampled value and hand-tuning for best eval performance. Specifically, we searched in the range $0.0025$ to $1$ for inner learning rate and $r = 5, 10$.

Table 2: C-SWM hits at rank 1 [4] and mean reciprocal rank accuracy with learning rate 1e-4 and 5e-4 on the Atari Pong (Pong) and Space Invaders (Space) datasets (mean and standard deviation over 5 trials). The numbers in the parenthesis after model names are the number of object slots $K$ and epochs.

|  | Configurations | 1 Step | | 5 Steps | | 10 Steps | |
|---|---|---|---|---|---|---|---|
|  |  | H@1 | MRR | H@1 | MRR | H@1 | MRR |
| Pong | 1e-4 (5, 500) | **63.6** $_{\pm1.6}$ | **75.8** $_{\pm1.2}$ | **26.1** $_{\pm1.7}$ | **43.1** $_{\pm1.7}$ | **16.1** $_{\pm1.4}$ | 28.9 $_{\pm2.4}$ |
|  | 1e-4 (5, 200) | 45.7 $_{\pm4.8}$ | 63.5 $_{\pm3.7}$ | 16.5 $_{\pm3.8}$ | 32.8 $_{\pm4.1}$ | 8.6 $_{\pm1.9}$ | 21.2 $_{\pm2.9}$ |
|  | 1e-4 (3, 500) | 52.5 $_{\pm6.8}$ | 68.6 $_{\pm4.4}$ | 24.0 $_{\pm3.5}$ | 40.7 $_{\pm4.0}$ | 11.8 $_{\pm3.3}$ | 25.1 $_{\pm4.7}$ |
|  | 1e-4 (3, 200) | 39.7 $_{\pm6.6}$ | 59.0 $_{\pm5.5}$ | 13.9 $_{\pm2.6}$ | 28.8 $_{\pm3.3}$ | 6.5 $_{\pm1.5}$ | 16.5 $_{\pm2.0}$ |
|  | 5e-4 (5, 500) | 56.8 $_{\pm9.6}$ | 71.6 $_{\pm6.4}$ | 21.2 $_{\pm4.9}$ | 37.6 $_{\pm5.3}$ | 13.3 $_{\pm3.9}$ | 26.5 $_{\pm5.0}$ |
|  | 5e-4 (5, 200) | 22.8 $_{\pm12.5}$ | 42.0 $_{\pm13.4}$ | 7.1 $_{\pm4.6}$ | 18.6 $_{\pm6.9}$ | 4.0 $_{\pm2.9}$ | 11.8 $_{\pm5.8}$ |
|  | 5e-4 (3, 500) | 29.7 $_{\pm8.6}$ | 49.4 $_{\pm8.1}$ | 20.4 $_{\pm8.2}$ | 37.0 $_{\pm9.3}$ | 14.4 $_{\pm5.4}$ | **30.1** $_{\pm7.2}$ |
|  | 5e-4 (3, 200) | 26.9 $_{\pm2.4}$ | 46.8 $_{\pm2.7}$ | 12.4 $_{\pm3.5}$ | 27.9 $_{\pm4.9}$ | 10.2 $_{\pm4.0}$ | 23.6 $_{\pm5.9}$ |
| Space | 1e-4 (5, 500) | 46.9 $_{\pm6.9}$ | 65.6 $_{\pm5.2}$ | 33.1 $_{\pm12.2}$ | 53.6 $_{\pm11.1}$ | 21.6 $_{\pm5.2}$ | 40.0 $_{\pm5.7}$ |
|  | 1e-4 (5, 200) | 37.1 $_{\pm8.0}$ | 56.8 $_{\pm7.3}$ | 19.6 $_{\pm9.3}$ | 38.1 $_{\pm12.2}$ | 10.0 $_{\pm3.5}$ | 23.5 $_{\pm4.9}$ |
|  | 1e-4 (3, 500) | **67.9** $_{\pm3.6}$ | **80.6** $_{\pm2.7}$ | **46.7** $_{\pm18.5}$ | **64.4** $_{\pm17.4}$ | **30.7** $_{\pm8.6}$ | **50.6** $_{\pm8.3}$ |
|  | 1e-4 (3, 200) | 54.1 $_{\pm7.6}$ | 70.6 $_{\pm5.7}$ | 30.3 $_{\pm13.1}$ | 50.4 $_{\pm14.9}$ | 16.8 $_{\pm3.1}$ | 33.4 $_{\pm4.0}$ |
|  | 5e-4 (5, 500) | 56.8 $_{\pm3.8}$ | 73.4 $_{\pm2.9}$ | 34.6 $_{\pm14.1}$ | 53.7 $_{\pm13.8}$ | 27.9 $_{\pm11.4}$ | 47.3 $_{\pm13.0}$ |
|  | 5e-4 (5, 200) | 40.0 $_{\pm9.0}$ | 60.8 $_{\pm7.4}$ | 14.2 $_{\pm5.8}$ | 31.1 $_{\pm7.5}$ | 11.5 $_{\pm4.0}$ | 26.0 $_{\pm6.2}$ |
|  | 5e-4 (3, 500) | 56.9 $_{\pm3.3}$ | 73.3 $_{\pm2.0}$ | 31.7 $_{\pm11.2}$ | 51.3 $_{10.5}$ | 23.9 $_{\pm14.0}$ | 42.3 $_{\pm13.8}$ |
|  | 5e-4 (3, 200) | 55.8 $_{\pm4.7}$ | 72.5 $_{\pm3.3}$ | 32.8 $_{\pm7.8}$ | 54.2 $_{\pm8.2}$ | 15.5 $_{\pm11.6}$ | 33.5 $_{\pm13.9}$ |

## F   Language Reasoning

We used the official CLUTRR-Baselines [4] code for all our experiments. For SRN, $H_{\text{agg}}$ is the same as in the circles experiment. The inner learning rate is set to $0.001$ and $r = 5$. For GRN, we use scene graph encoder based on [2] as $H_{\text{agg}}$. The inner learning rate is set to $0.0005$ and $r = 10$. The hyper-parameters are selected by starting with a random sampled value and hand-tuning for best eval performance. Specifically, we searched in the range 1e-3 to 1e-5 for l2-penalty, by which the result is affected the most.

## G   Computing Infrastructure and Runtime

We use PyTorch [6] for all of our experiments. All experiments were run on a SLURM cluster - primarily on a machine with a single RTX 2080TI Nvidia GPU. All runtime results are reported with those.

The effect that SRN has on runtime is dependent on the size of the perceptual end as well as the relational reasoning end. However, the effect that SRN has on runtime is disproportionate to the actual complexity of the models used in SRN, as the inner optimization loop means that SRN must execute the models within it multiple times.

In the CLUTTR and circle reconstruction experiment, for example, the runtime is dominated by the perceptual end as well as the image reconstruction. As such, SRN increases the time per epoch by 10%.

On the other hand, for Sort-of-CLEVR and C-SWM, our experiments used a SRN module that increases runtime by about 2-3x. We note that although this is a significant runtime increase, it is somewhat ameliorated by 2 factors: 1. SRN tends to increases convergence rate. For example, on Sort-of-CLEVR SRN reaches 99% performance on non-relational questions at epoch 3 vs epoch 10 (final performance for both is 100%). 2. We have not focused on reducing the runtime of SRN. As we have reused the same FSPool set encoder with the same parameters for most all of our tasks, preliminary experiments suggest that we can improve runtime significantly by reducing the size of this encoder, possibly without affecting performance. Similarly, we can reduce runtime significantly by decreasing the number of inner optimization steps taken.

## Footnotes

[2]https://github.com/kimhc6028/relational-networks/tree/74cd9ac0703db01c6268a6515015b98bed3e4602

[3]https://github.com/tkipf/c-swm

[4] https://github.com/koustuvsinha/clutrr-baselines

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

Figure 3: Ten sampled reconstruction and decomposition results. From left to right, column-wise: original images, SRN reconstruction, SRN decomposition, baseline reconstruction, baseline decomposition.

Figure 4: Five sampled reconstruction and decomposition results. From left to right, column-wise: ground truth objects image, SRN reconstruction, SRN decomposition, baseline reconstruction, baseline decomposition

Figure 5: Five sampled reconstruction and decomposition results. From left to right, column-wise: ground truth objects image, SRN reconstruction, SRN decomposition, baseline reconstruction, baseline decomposition

(a) Ground Truth Image       (b) SRN Reconstruction       (c) Baseline Reconstruction

Figure 6: The example ground truth and reconstructions for Figure 7 and 8.

Figure 7: The decomposition of SRN for example in Figure 6.

Figure 8: The decomposition of baseline for example in Figure 6.

Figure 9: Examples of Robustness (Synthetic). The left one is the original configuration, and the right one is an example of how rotation looks like.