[Reviews · NeurIPS 2020]

Review 1

Summary and Contributions: This paper presents a method for representing a real valued input vector as an unordered set of 2 or more real valued vectors which represent most of the relevant information in the original input vector. The representations are trained end-to-end using standard training losses for either label prediction or regeneration. Their method is simply added as an additional neural network layer in a standard training setup, where unordered object representations are expected to be useful. The basic idea is similar to energy based models. The model first directly trains a flat vector representation of data (using the end-to-end training objective). At test time an iterative gradient descent procedure is performed to regress the set representation towards values that can be used to generate the flat representation. At training time this gradient descent procedure is unrolled and back propagation is used to train the whole procedure end-to-end. They provide results showing that adding their layer improves baseline models on Sort-of-CLEVR, next world state prediction for RL, and relational reasoning in natural language.

Strengths: - novel method for generating a set representation of data - results showing that the generated set representations generalize better than their contiguous counterparts - show improvement across three different diverse tasks rather than one specific task - their method requires only on additional layer and so it can be directly applied to many other models.

Weaknesses: - They only compare to relation networks and not to FiLM on the Sort-of-Clevr dataset which produces better results - They agreed to include the FiLM results in the final paper which slightly edges out their results on the "Furthest" Sort-of-CLEVR dataset (96.1%) which is slightly disappointing since was whether they showed the biggest gain over RN on it's own.

Correctness: - The claims and method are correct as far as I can tell

Clarity: - In general the paper was clear and understandable

Relation to Prior Work: - I am not an expert in this area, but I believe their method is novel | - The paper does not do a very good job of drawing the connection to other prior work on iterated methods such as Energy Based Models, recurrent attention models (such as Attend, Infer, Repeat), Capsule Networks, soft clustering, etc. - If the paper is accepted, the authors are also encouraged to discuss the relationship with: https://arxiv.org/pdf/2006.15055.pdf which was published after the Neurips submission.

Reproducibility: Yes

Additional Feedback: The broader impact statement mostly rehashes the motivations for the work. My understanding is that this should focus more on the societal level and ethical implications which are mostly glossed over in their statement. Thank you for your feedback, you have clarified my confusion and I have update my review accordingly.


Review 2

Summary and Contributions: This paper proposes an iterative procedure that mitigates the responsibility problem that occurs during the learning of latent representation of sets of objects in the context of relational learning with neural networks. In contrast with previous approaches that target the responsibility problem, this technique can be applied to generic downstream relational tasks over raw data without explicit supervision. The proposed module, dubbed Set Refiner Network (SRN), can be plugged into any end-to-end relational neural model in order to improve the latent set representations that are learned in downstream tasks.

Strengths: The contribution is very relevant and the proposed approach can benefit a large number of relational reasoning tasks. The empirical evaluation seems exhaustive and shows the effectiveness of SRN.

Weaknesses: I wish that the paper gave a more intuitive explanation of why the proposed technique is effective in mitigating the responsibility problem.

Correctness: The empirical evaluation looks solid. The methodology seems correct although I'm not an expert in this area.

Clarity: I think that the paper is well written.

Relation to Prior Work: Yes.

Reproducibility: Yes

Additional Feedback: The contribution is relevant for the community and the paper is well written. I particularly enjoyed the extensive empirical evaluation, although I am not an expert in this area and I can't really tell whether the comparisons lack some alternative approaches or if the baselines are adeguate. First, the authors show how using SRN helps in learning more meaningful set representations with respect to a (in my opinion, reasonable) baseline approach. Second, experiments on different relational reasoning tasks show a significant improvement when SRN is plugged into the reasoning pipeline. While I think that I got an intuitive idea, I wish that the paper put more focus on *why* the proposed iterative procedure is helpful in learning effective set representations, possibly including images that help the reader to better understand the concept. Minors: Line 124: a data embedding --- I thank the authors for the feedback. It didn't drastically change my opinion on this work, I confim my previous score.


Review 3

Summary and Contributions: This paper introduces the Set Refiner Network (SRN): a module based on the Deep Set Prediction Network (DSPN; Zhang et al., 2019) for predicting set representations. The SRN is obtained by replacing the supervised classification loss in DSPN with a task-specific module/loss, such as an image reconstruction module/loss. SRN is tested in unsupervised object detection via image reconstruction, visual reasoning on Sort-of-CLEVR and in modeling of Atari environments. The experimental results indicate that SRN can learn set-structured intermediate representations that benefit downstream task performance, compared to ablation-based baselines that do not make use of the SRN module.

Strengths: This work is potentially of high interest to the NeurIPS community as it empirically demonstrates how DSPN can be employed in conjunction with downstream tasks other than supervised set prediction. The considered downstream tasks are of high relevance, despite their toy-like scale, and the proposed approach shows empirical benefits when used in conjunction with existing architectures. The theoretical claim in Theorem 1 seems sound, although it is not entirely clear how realistic/relevant the assumptions are that go into Theorem 1 (see “additional feedback” below). The empirical evaluation is diverse (in terms of application domains) and insightful. The experiments on robustness of the proposed approach (compared to a reasonable baseline) are interesting and insightful.

Weaknesses: In terms of the proposed method, there is little novelty over DSPN (Zhang et al., 2019): SRN is largely a re-branding of DSPN. Indeed, the attached code implementation of SRN simply imports and applies the DSPN module. This is not necessarily a bad thing: it is perfectly fine to re-use prior work, but I think that giving an existing module a new name leads to unnecessary confusion. There is no experimental comparison to related works that propose iterative models along similar lines of reasoning (e.g. IODINE [Greff et al., 2019] or Relational Deep Reinforcement Learning [Zambaldi et al., 2018]). All baselines are the same form of model ablation (ablation of the SRN network). It would be interesting to compare to existing models that obtain set-structured hidden representations, such as IODINE (for the image domain). For visual reasoning and modeling of environments, an architecture such as the one presented in Relational Deep Reinforcement Learning (Zambaldi et al., 2018), which uses recurrent self-attention on top of a CNN, would be a well-suited baseline. At the current stage, it is unclear what benefits SRNs provide over these existing approaches, and an explicit comparison would make the paper stronger. Ablation studies: the SRN module itself consists of a number of components and hyperparameters. It would be good to analyse these choices in detail via ablation studies. For example: how important is the FSPool layer as opposed to some other choice of pooling (e.g. average pool), how important is it to tune the inner vs outer learning rate, how important is the size of the MLP before the the set pooling layer, etc. Answering these questions explicitly would make the paper stronger. UPDATE: Some of my comments were addressed and I think that the paper could qualify for acceptance despite remaining weaknesses (especially with respect to novelty over DSPN and with respect to baseline comparisons and ablation studies). I have updated my recommendation to weak accept (6).

Correctness: Overall, the methodology and the theoretical analysis appears to be correct. The method makes sense and the empirical analysis is sound. Some of the claims, however, are debatable, such as: 1) “We are the first to demonstrate that [...] using iterative inference resolves a fundamental issue in relational reasoning over raw data” -- I think both the “Relational Deep Reinforcement Learning” paper by Zambaldi et al. (2018) and the “Relational recurrent neural networks” paper by Santoro et al. (2018) argued in a similar manner. 2) Referring to the experimental setting from the C-SWM paper as “reinforcement learning” is potentially misleading: no reward signal is used in these experiments. 3) The authors claim “substantially” improved predictions also for the comparison to C-SWM. Looking at the reported numbers, however, the improvements don’t seem to be very significant given the larger variance for results using the SRN module. 4) “Despite this assumption, typical set generation process still fail to enforce permutation invariance, which is fixed by including our SRN” → This statement is unclear. SRN is used on initial set representations that are ordered in a deterministic way, and hence the overall predicted set is deterministically ordered as all updates by SRN are permutation equivariant and deterministic. It is unclear how this relates to the statement of “permutation invariance” in this particular claim.

Clarity: Overall, the paper is well written and easy to follow, but contains a number of typos.

Relation to Prior Work: Apart from the two papers mentioned above [“Relational Deep Reinforcement Learning” by Zambaldi et al. (2018) and “Relational recurrent neural networks” by Santoro et al. (2018)], related work appears to be discussed at an adequate level.

Reproducibility: Yes

Additional Feedback: It is not entirely clear to me how realistic the assumptions for Theorem 1 in the context of the considered tasks are: Couldn’t we just avoid the responsibility problem by assuming that the image is generated from an ordered list of entities (e.g. ordered by location or other properties)? An example for why the responsibility problem might be a non-issue is the MONet model (Burgess et al., 2019): it assumes a fixed ordering of outputs and is not permutation-equivariant, yet it can produce near perfect object decompositions of complex scenes.

[Author Response · NeurIPS 2020]

We thank the reviewers for taking the time to read our submission and offer feedback. We would like to emphasize that all reviewers found several positive points of the paper. For example, R1 found the method simple and novel, R2 said that our approach "benefits a large number of relational reasoning tasks," and R3 noted that the empirical evaluation is "diverse and insightful." Below, we discuss what we found to be the three main concerns raised by the reviewers: (1) comparisons to different approaches; (2) clarity of Theorem 1, related assumptions, and relation to SRN; and (3) novelty compared to DSPN. We believe that we can address these issues for a strong camera-ready paper.

**(1) Comparisons to different approaches.** *R1 discussed FiLM as an alternative.* Since FiLM does not explicitly generate a set, it is difficult to compare against for set generation (although, combining the two is an interesting avenue for future research). Still, we tested FiLM using the same convolution backbone and it achieved similar performance on the furthest task (96.1%), but did much worse on the closest task (91.5%). We will include these results.

*R3 discussed Relational Deep RL (RDRL).* We want to emphasize that RDRL is iterative across *time steps* and is not using iterative inference to refine a set representation. Thus, in the context of our paper, RDRL has a conceptually similar role to our baseline Relational Network (RN), as they both start by assuming that they have a set of entities. Because of this, we expect RDRL to have the same issues as RNs and would also be a *beneficiary* of the SRN module.

*R3 also discussed IODINE.* IODINE is an unsupervised object detection method, which has limitations as it is divorced from the downstream task. For this reason, these types of methods are typically not used with the models we consider in our paper (e.g., Relational Networks or C-SWM). For example, the C-SWM paper argues that "typical failure modes [of using methods like IODINE] include ignoring visually small, but relevant features..., or wasting model capacity on visually rich, but otherwise potentially irrelevant features." In addition, methods like IODINE are sensitive to extraneous entities, as it is unaware which entities are relevant to the task at hand. On the other hand, SRN is robust to this issue, as shown by the "cluttered background" experiments in the Appendix. Finally, IODINE is only applicable to images, so we cannot run it on CLUTTR. All that said, we have tried but failed to obtain reasonable results from IODINE during the rebuttal period, largely due to the significant compute required for IODINE.

*R3 was also concerned with improvements over C-SWM.* As all runs on this task have large epoch-to-epoch variance, we ran updated experiments to reduce variance by selecting the best epoch for each run based on H@5, and use 10 runs. For both experiments, all of our improvements on 5/10 steps are significant, and we will include statistical tests in the paper. For example, on Space Invaders (3 objects, H@1, 10 steps), where SRN performance is most variable, a Welch's t-test provides a 95% confidence interval of 3.7% to 15.7% improvement.

**(2) Clarifications on Theorem 1 and how SRN mitigates the responsibility problem.** *R3 raised concerns on how realistic the assumptions are, noting MoNet.* This is a subtle point, and we thank the reviewer for raising it. Theorem 1 still applies as long it is *possible* to generate the data from a set continuously, even if there is *some* map that generates the data from an ordered list. Thus, the assumptions are broader than they may appear in the text. A method that uses a canonical ordering for a set (as in MoNet) still needs to reorder the entities discontinuously for some continuous transformation of the objects. We will clarify these ideas and issues in the text surrounding Theorem 1.

*R2 and R3 were looking for more intuition on why SRN is effective.* The main intuition, as we illustrate in Figure 1, is that SRN is able to model the discontinuity required to handle the responsibility problem (similar to DSPN). Adding the inner optimization loop to a feedforward network enables us to model this discontinuity. In the language of Theorem 1, this means that SRN can model the discontinuous function $h$ for the permutation-invariant map $h \circ g$. To help illustrate this point, we can flesh out the numerical example that we used to generate parts of Figure 1.

*R1 had a concern with the proof of Theorem 1.* This is a misunderstanding. The output of $h$ is an element of $\mathbb{R}^{n \times d}$, which is equivalent to a "list." Thus, $h \circ g$ is indeed a function from sets to lists. We can clarify this.

**(3) Novelty compared to DSPN.** *R3 had concerns on framing contributions and naming, with respect to DSPN.* We agree that the primary novelty is demonstrating how resolving the responsibility problem can improve performance and robustness in tasks that use latent set structure (and not in the inner optimization loop itself, which uses similar ideas to energy-based models, representation inversion, and DSPN). Additionally, we think there is a slight misunderstanding here. Although we started from the DSPN source code, we have made significant changes for our setting. Methodologically, SRN removes DSPN's dependency on ground truth sets during training. From the DSPN paper: "when naively using [the image embedding] as input... our decoding process is unable to predict sets correctly from it. To fix this, we add a term to the loss [that depends on ground truth sets]". In other words, DSPN cannot be used in settings other than supervised set prediction. SRN addresses this limitation by refining a predicted set instead of using a shared initialization across all inputs, which also allows easy integration into existing relational reasoning pipelines. Since the goals and methodology are fundamentally different, we think that a different name is appropriate; however, we can think of a name that more accurately reflects the connections.

[Meta-Review · NeurIPS 2020]

All the reviewers agreed that this paper should be accepted. The authors are encouraged to discuss the remaining issues (discussing the novelty over DSPN and address add some more baseline comparisons and ablation studies R3) in the camera-ready.